# Expanding the Study of the Cytotoxicity of Incomptines A and B against Leukemia Cells

**DOI:** 10.3390/molecules27051687

**Published:** 2022-03-04

**Authors:** Fernando Calzada, Normand Garcia-Hernandez, Sergio Hidalgo-Figueroa, Elihú Bautista, Elizabeth Barbosa, Claudia Velázquez, Marta Elena Hernández-Caballero

**Affiliations:** 1Unidad de Investigación Médica en Farmacología, Unidad Médica de Alta Especialidad, Hospital de Especialidades-2° Piso CORSE Centro Médico Nacional Siglo XXI, Instituto Mexicano del Seguro Social, Av. Cuauhtémoc 330, Col. Doctores, Ciudad de México 06725, Mexico; 2Unidad de Investigación Médica en Genética Humana, Unidad Médica de Alta Especialidad, Hospital de Pediatría, Centro Médico Nacional Siglo XXI, Instituto Mexicano del Seguro Social, Av. Cuauhtémoc 330, Col. Doctores, Ciudad de México 06725, Mexico; 3CONACyT-Instituto Potosino de Investigación Científica y Tecnológica A. C., Camino a la Presa San José 2055, Lomas 4ª Sección, San Luis Potosí 78216, Mexico; sergio.hidalgo@ipicyt.edu.mx (S.H.-F.); francisco.bautista@ipicyt.edu.mx (E.B.); 4Sección de Estudios de Posgrado e Investigación, Escuela Superior de Medicina, Instituto Politécnico Nacional, Salvador Díaz Mirón esq. Plan de San Luis S/N, Miguel Hidalgo, Casco de Santo Tomas, Ciudad de México 11340, Mexico; rebc78@yahoo.com.mx; 5Área Académica de Farmacia, Instituto de Ciencias de la Salud, Universidad Autónoma del Estado de Hidalgo, Km 4.5, Carretera Pachuca-Tulancingo, Unidad Universitaria, Pachuca 42076, Mexico; cvg09@yahoo.com; 6Facultad de Medicina, Biomedicina, Benemérita Universidad Autónoma de Puebla, Puebla 72410, Mexico; ehdezc@yahoo.com

**Keywords:** incomptine A, incomptine B, sesquiterpene lactones, cytotoxic activity, human cancer cell lines, leukemia, docking

## Abstract

Heliangolide-type sesquiterpene lactones (HTSLs) are phytocompounds with several pharmacological activities including cytotoxic and antitumor activity. Both bioactivities are related to an α-methylene-γ-lactone moiety and an ester group on carbon C-8 in the sesquiterpene lactone (SL) structure. Two HTSLs, incomptines A (**AI**) and B (**IB**) isolated from *Decachaeta incompta*, were evaluated for their cytotoxic activity on three leukemia cell lines: HL-60, K-562, and REH cells. Both compounds were subjected to a molecular docking study using target proteins associated with cancer such as topoisomerase IIα, topoisomerase IIβ, dihydrofolate reductase, methylenetetrahydrofolate dehydrogenase, and Bcl-2-related protein A1. Results show that **IA** and **IB** exhibit cytotoxic activity against all cell lines used. The CC_50_ value of **IA** was 2–4-fold less than etoposide and methotrexate, two anticancer drugs used as positive controls. The cytotoxic activity of **IB** was close to that of etoposide and methotrexate. The molecular docking analysis showed that **IA** and **IB** have important interaction on all targets used. These findings suggest that **IA** and **IB** may serve as scaffolds for the development of new treatments for different types of leukemia.

## 1. Introduction

Medicinal plants are a renewable source for phytopharmaceutical agents with important potential to treat several diseases such as cancer [1,2,3,4]. In this sense, research into vegetal extracts of medicinal plants has led to the discovery of anticancer drugs including parthenolide, etoposide, taxol, and vincristine [5,6,7,8]. Plants contain several bioactive secondary metabolites including alkaloids, flavonoids, phenols, sterols, quinones, and terpenoids. Among the terpenoids, the HTSLs are a large group of secondary metabolites with a 15-carbon skeleton and a low molecular weight, and contain oxygenated groups such as alcohols, ketones, aldehydes, epoxides, acids, and/or an *a*-methylene-γ-lactone. Sesquiterpene lactones (SLs) are secondary metabolites that accumulate in the leaves, roots, and stem of plants belonging to diverse Asteraceae genera including *Artemisia, Arnica, Tanacetum*, and *Decachaeta*. These compounds have several biological properties such as antimicrobial, anti-inflammatory, cytotoxic, and anti-tumor [8,9]. In general, the biological properties of SLs like cytotoxic and antitumor agents have been associated with the presence of an *a*-methylene-γ-lactone in their structure; this moiety has the capacity to act as a Michel acceptor and react with sulfhydryl residues of proteins. In addition, SLs exercise their anticancer properties by changing the redox cell balance, as well as reducing glutathione depletion, preventing NF-kB activation, and increasing intra-cellular reactive oxygen species levels. In addition, they can inhibit glycolytic enzymes and downregulate Bcl-2 antiapoptotic proteins [8,9].

*Decachaeta incompta* (DC) R. M. King and H. Robinson (Figure 1) belongs to the Asteraceae family in the Asterales order; it is an erect subshrub that grows up to 3 m tall. The plant has glands in the leaves, reddish or yellowish corollas and styles, straight stems, and alternate and ovate leaves. It has white florets that are gradually more densely reddish or yellowish glandular distally. The plant is native to Guatemala and to Mexican states such as Oaxaca, Michoacan, Jalisco, Puebla, and Veracruz [10]. The aerial parts of this species are used in Oaxacan traditional medicine to treat diarrhea. Bioassay-guided fractionation of dichloromethane extraction of the aerial parts of *D. incompta* led to isolation of four heliangolide-type SLs named incomptines A–D. Regarding their biological properties, they have been reported to possess antiamoebic, antigiardial, antibacterial, antipropulsive, trypanocidal, phytotoxic, spermatic, cytotoxic, and antitumor effects. In this sense, we hypothesized that the antiprotozoal and phytotoxic activities of these SLs are associated with the presence of an acetate moiety at C-8 of the germacrane framework. In contrast, the antipropulsive and antibacterial properties may be associated with the presence of a hydroxy group at the same position of this backbone [11,12]. 

Leukemia is a cancer of the blood-forming tissues characterized by an increase in uncontrolled growth of immature white blood cells or leukocytes in the blood, spleen, and bone marrow. There are many classes of leukemia; it is classified in agreement to the course of disease and the dominant type of white blood cell involved in the disease. Examples include acute myeloid leukemia, acute lymphocytic leukemia, chronic myeloid leukemia, and chronic lymphocytic leukemia. Symptomatic patients present anemia, fever, bleeding, bone pain, tenderness, fatigue, weakness, excessive sweating, headaches, nausea, and swelling of the lymph nodes. In 2018, around 430,000 cases worldwide were reported with leukemia, which accounted for 300,000 deaths. The anticancer drugs commonly used, alone or combined, for treating leukemias include cyclophosphamide, fludarabine, prednisone, chlorambucil, etoposide, methotrexate, and doxorubicin. All these drugs present significant side effects, in some cases multidrug resistance, and in the case of etoposide or methotrexate secondary leukemia may develop. In an effort to improve cancer therapy, it was proposed to employ secondary metabolites isolated out of Asteraceae plants for developing new treatments against malignancies, including non-solid tumors such as leukemias [8,13,14,15,16]. 

This study forms part of our research on the cytotoxic properties of incomptines A (**IA**) and B (**IB**). In this work, the cytotoxic activity of **IA** and **IB** was evaluated against three cell lines causing leukemia (HL-60, K-562, and REH). In addition, a molecular docking study was carried out to understand the potential mechanism of action on five molecular targets involved in survival and proliferation of cancer cells: topoisomerase IIα (TIIα), topoisomerase IIβ (TIIβ) dihydrofolate reductase (DHFR), methylenetetrahydrofolate dehydrogenase (MTHFD), and Bcl-2-related protein A1 (BCL-2). 

## 2. Results

### 2.1. Isolation and Cytotoxic Activity of Incomptine A (**IA**) and Incomptine B (**IB**) 

In the present study, the aerial parts of *D. incompta*, collected in the State of Oaxaca, Mexico, were extracted exhaustively with dichloromethane. An amount of 2.12 g of brown extract was yielded, which had a pasty consistency (DCE, 7.85%). The DCE from *D. incompta* was purified by column chromatography to give two germacrane-type sesquiterpene lactones named incomptine A **(IA**) and incomptine B (**IB**) **2** (Figure 2); both compounds were identified through comparison of retention times in HPLC-DAD (Figure 3 and Figure 4) and NMR spectra. 

The cytotoxic activity of the incomptines A (**IA**) and B (**IB**) were tested on three leukemia cell lines (HL-60, K-562, and REH) and one lymphoma cell line (U-937) by MTT assay. Lymphoma cell line (U-937) was used as a positive control considering that incomptine A has been demonstrated to exhibit strong cytotoxic activity on this cell line. To compare the cytotoxic effects of the HTSLs analyzed in this work, incomptines A (**IA**) and B (**IB**), we included etoposide (**ET**) and methotrexate (**MTX**) as positive controls, i.e., two anti-leukemia drugs. The cytotoxicity assays revealed that both heliangolide-type sesquiterpene lactones have cytotoxic activity against all assayed leukemia and lymphoma cell lines (Table 1). The most potent compound was the incomptine A (**IA**) with CC_50_ values from 0.3, 0.6, 0.3, and 0.4 µM on U-937, HL-60 K-562, and REH cell lines, respectively. In all the cases, their cytotoxic activities were better than **ET** (1.2, 1.4, 0.7, and 1.1 µM, respectively) and **MTX** (1.5, 0.65, 3.4, and 2.7 µM, respectively). Incomptine B (**IB**) was less potent, displaying CC_50_ values from 1.9, 1.0, 1.9, and 2.1 µM, respectively. Nevertheless, its cytotoxic activities were close to etoposide and methotrexate. All compounds tested exhibited dose-dependent cytotoxic effects (Figure 5, Figure 6, Figure 7 and Figure 8) on all cell lines used. The results revealed that all leukemia and lymphoma cell lines used were sensitive to the cytotoxic effects of the tested HTSLs.

Although the data are limited, a structure-cytotoxicity relationship was established from the CC_50_ values in the tested leukemia cells and revealed that the presence of an acetate moiety at C-8 favors the cytotoxicity, as occurs for antiprotozoal and phytotoxic activities [11,12].

### 2.2. Molecular Docking Studies of Incomptine A (**IA**) and Incomptine B (**IB**) on Five Selected Pharmacological Receptors Associated to Cancer

Considering previous in vitro results and the targets reported for the two anti-leukemia drugs used as positive controls, we decided to study five potential pharmacological receptors associated to cancer: topoisomerase IIα (TIIα), topoisomerase IIβ(TIIβ) dihydrofolate reductase (DHFR), methylenetetrahydrofolate dehydrogenase (MTHFD), and Bcl-2-related protein A1 (BCL-2) [17,18,19]. Residues of interaction between incomptines A (**IA**) and B (**IB**), methotrexate (**MTX**), and etoposide (**ET**) against the five molecular targets (Figure 9, Figure 10 and Figure 11; Table 2) showed that incomptine B (**IB**) had greater affinity on the proteins TIIαDHFR and THFD with ΔG values of −7.4 kcal/mol, −8.1 kcal/mol, and −7.9 kcal/mol, respectively. In contrast, incomptine A (**IA**) exhibited the best affinity on the protein TPIIβ, with a ΔG value of −6.5 kcal/mol. Both incomptines showed the same affinity to BCL2 protein with a ΔG value of −7.3 kcal/mol. However, incomptines A (**IA**) and B (**IB**) exhibited less affinity than **MTX** and **ET** (Figure 11).

The specific interactions of incomptine A (**IA**) with all targets are displayed in Figure 9. Initially, the interaction of **IA** with human topoisomerase IIα is formed by two hydrogen bonds. First, the amine group of deoxyadenosine (DA12) bonds as a donor with the carbonyl group of the acetoxy; the second bond is between the amine group of deoxycytidine (DC11) as a donor and the oxygen atom of the lactone (Figure 9A,B). Subsequently, the interaction with human topoisomerase IIβ is formed by only one hydrogen bond, between the amine group of deoxyguanosine (DG13) as a donor and the same carbonyl group of the acetoxy (Figure 9C,D). The binding of **IA** with the dihydrofolate reductase (DHFR) is formed by two hydrogen bonds (H bonds); one of these bonds is between the oxygen atom of the epoxide group with Ala9, and the second non-conventional interaction (carbon hydrogen bond) is between the carbonyl group of the acetoxy moiety and the Gly116 residue (Figure 9E,F). Additionally, **IA** establishes three hydrogen bonds with methylenetetrahydrofolate dehydrogenase (MTHFD); two of these bonds are between the carbonyl group and oxygen atom of lactone with Gln100, and the last one is between Leu101 as a donor and the oxygen atom of the carbonyl group as the acceptor (Figure 9G,H). Finally, the binding of **IA** with the B-cell lymphoma 2 protein (BCL2) is by two hydrogen bonds between the amide group of the Trp141 and Gly142 residues (the donor) and the oxygen atom of the hydroxyl group is the acceptor (Figure 9I,J). The interactions of incomptine B (**IB**) with all targets are displayed in Figure 7. However, the acetoxy group is not present in **IB** and the interaction with human topoisomerase IIα (TIIα) is formed by two hydrogen bonds. The amine group of Gly488 bonds as an acceptor and the hydroxyl group; the second bond is between the amine group of deoxyguanosine (DG13) as a donor and the oxygen atom (carbonyl group) of the lactone (Figure 10A,B). Subsequently, the interaction of **IB** with human topoisomerase IIβ(TIIβ) is formed by only one hydrogen bond, between the amine group of deoxyadenosine (DA12) as a donor and the carbonyl group of the lactone (Figure 10C,D). The interaction of **IB** with dihydrofolate reductase (DHFR) established two hydrogen bonds; one of these bonds is between the oxygen atom of the carbonyl group of lactone with Ala9, and the second hydrogen bond is between the hydroxyl group and the Val115 residue (Figure 10E,F). In addition, **IB** establishes four hydrogen bonds with methylenetetrahydrofolate dehydrogenase (MTHFD); two of these bonds are between the carbonyl group of lactone with Gln100 and Lys56, and the last two are between Leu101 as a donor and acceptor, with the hydroxyl group as the donor (Figure 10G,H). Finally, **IB** missed polar interactions with any residues of the B-cell lymphoma 2 protein (BCL2, Figure 10I,J). 

## 3. Discussion

Leukemia is a cancer that is present worldwide and affects people of all ages, although predominantly in childhood. In 2018, nearly 430,000 people were diagnosed with any type of leukemia, which accounted for 300,000 deaths around the world [13,14].

In this context, there has been renewed interest during the last years in phytochemicals as potential chemopreventive and chemotherapeutic agents against cancer. These secondary metabolites include flavonoids, alkaloids, acetogenins, and terpenoids. In the latter group, sesquiterpene lactones (SLs) have received considerable attention during the last 17 years [8,9,20,21,22]. 

Here, we have reported the cytotoxic activity and docking analysis on five molecular targets relevant for cancer treatment of the two major heliangolide-type sesquiterpene lactones from *Decachaeta incompta*. Incomptine A (**IA**) exhibited the best cytotoxic activity; its effects were better than etoposide (**ET**) and methotrexate (**MTX**), two antitumor agents used currently for treating cancer, which were used as positive controls. Incomptine A showed a dose-dependent cytotoxic effect (Figure 5) on all cell lines used. These results suggest that **IA** could be considered promising antitumor compound. However, although the data are limited, the structure-effect correlation revealed that cytotoxic activity on U-937, HL-60, K-562, and REH human cell lines seems to be related to the presence of an 8-acetyl group of the heliangolide framework. Incomptine B (**IB**) that has a free hydroxyl at C-8 exhibited less cytotoxic activity. In addition, it is in agreement with the observed antiprotozoal and phytotoxic activity [11,12]. Notwithstanding that **IB** was less potent than **IA**, its cytotoxic activity was close to that of etoposide and methotrexate, suggesting it is also a good candidate for the development of new anticancer drugs. Further studies are needed in order to elucidate the mechanism of action of these compounds and to evaluate their bioavailabilities [17,23,24]. To our knowledge, this is the first report of comparative cytotoxic properties of incomptines A (**IA**) and B (**IB**) on leukemia human cell lines HL-60, K-562, and REH. It is important to point out that the biological properties of SLs are associated with alkylation of nucleophiles through their a-methylene-γ-lactone moiety. This moiety can react readily to form adducts with nucleophiles such as sulfhydryl groups or free thiols by Michel-type addition [23]. In this sense, cysteine residues in proteins, as well as the free intracellular GSH, are the main targets of SLs. These interactions cause reduction or inhibition of enzyme activity or disruption of GSH metabolism and the vitally important intracellular cell redox balance. In addition, extensive research suggests that SLs exercise their antitumor effects by reacting on proteins and enzymes or interfering with some key biological processes, including the sarco/endoplasmatic reticulum calcium ATPase pump, proteases, transferrin receptors, nuclear factor-kappa B, and E3 ubiquitin-protein ligase Mdm2, as well as the p53 gene, angiogenesis, and metastasis in cancer cells [8,9]. In this context, parthenolide (**PTL**), a sesquiterpene lactone isolate of *Tanacetum parthenium*, has the a-methylene-γ-lactone moiety; it has shown potent anticancer activity and is currently being tested in cancer clinical trials. PTL is the first small molecule found to be selective on cancer stem cell lines. In vitro and in vivo anticancer properties of **PTL** can be associated to its important inhibition of nuclear factor kappa B (NF-kB), which is aberrantly and stably activated in various cancers. Other SLs that possess an a-methylene-γ-lactone moiety and demonstrated significant anticancer potential include eupatolide, deoxyelephantopin, and dehydrocostus lactone [25,26,27]. 

In relation to docking studies, incomptines A and B showed similar energy of interaction values (ΔG) but these were slightly lower than the control drugs. However, it is important that the acetoxy group of incomptine interacts as an acceptor group, and in this position, it is desirable to interact with all targets, indicating an increase in recognition with the proteins involved in cancer. These results correlate with in vitro activity studies. In addition, the molecular docking analysis suggests that cytotoxic activity of **IA** and **IB** against all leukemia cell lines used in this study may be associated with the effects on the five targets used: human topoisomerase IIα, human topoisomerase IIβ, human dihydrofolate reductase, human methylenetetrahydrofolate dehydrogenase, and human B-cell lymphoma 2 protein. 

## 4. Materials and Methods

### 4.1. Collection and Identification of Decachaeta Incompta

*Decachaeta incompta* (D.C.) King and Robinson (Asteraceae) aerial parts were collected in Portillo Nejapa de Madero (16°36′00″ N, 95°59′00″ O) State of Oaxaca, Mexico. The plant material was authenticated, and a voucher specimen (Aguilar, 15311) was deposited at the medicinal Herbarium IMSSM of the Instituto Mexicano del Seguro Social (IMSS).

#### 4.1.1. Chemicals

Methotrexate, etoposide, 3-(4,5-dimethylthiazol-2-yl)-2,5-diphenyltetrazolium bromide (MTT), dimethyl sulfoxide (DMSO), L-glutamine, penicillin/streptomycin, RMPI 1640 medium, acetonitrile, methanol, and acetic acid HPLC grade were purchased from Sigma-Aldrich, USA. Ethanol, dichloromethane, hexane, and methanol AR grade were purchased from JT Baker, Mexico. Fetal bovine serum was purchased from Gibco, Mexico.

#### 4.1.2. Preparation of the Aerial Parts Extract

Air-dried plant material (25 g) was ground and extracted by percolation at room temperature with dichloromethane (350 mL). After filtration, the solvent was evaporated under vacuum to yield 2 g of extract of brown color and a pasty consistency.

#### 4.1.3. Isolation and Purification of Germacrane-Type Sesquiterpene Lactones, Incomptines A (**IA**) and B (I**B**)

The dichloromethane extract (1.8 g) was subjected to column chromatography (CC) over silica gel (20 g, 70-230 mesh, Merck, Darmstadt, Germany) using hexane and a mixture of dichloromethane-MeOH (7:3–5:5) to give five fractions (Fr1–Fr5). Fractions 4 and 5 were combined and resolved by CC over silica gel (20 g) using a mixture of solvents; dichloromethane in methanol (7:3–5:5) to yield incomptine A (95 mg) and incomptine B (499 mg). Incomptines A (**IA**) and B (**IB**) were identified by comparison of their retention times (RTs) in HPLC-DAD (Figure 3) and NMR-H^1^ and C^13^ using authentic sample disposables in our laboratory [11,17]. Incomptine A: white crystals, mp 175–176 °C, RT 45.31 min. Incomptine B: whyte crystals, mp 178–179 °C, RT 40.27 min.

#### 4.1.4. HPLC-DAD Analysis 

Dichloromethane extract (100 mg) or incomptines A (**IA**, 2 mg) and B (**IB**, 2 mg) were dissolved in methanol (10 mL or 2 mL, respectively) and 20 µL of the sample was injected to HPLC. HPLC separations were performed on a Waters 2795 Alliance equipped with a Waters 996 detector photodiode array collecting data at 240 nm; a column Waters Spherisorb S5 ODS2 (4.5 × 250 mm, 5 µm) was used with a logarithmic gradient from 96% of aqueous acetic acid 2% and 4% of CH_3_CN to 50% of aqueous acetic acid 2% and 96% of CH_3_CN over a period of 60 min at a flow rate of 1 mL min^−1^. All the solvents were HPLC grade.

### 4.2. Assay for Growth Inhibition

#### 4.2.1. Leukemia Cell Lines

The human pro-monocyte myeloid leukemia U-937 (ATCC: CRL 1593.2, Middlesex, UK), the human acute myeloid leukemia HL-60 (ATCC: CRL 3306, Middlesex, UK), the human chronic myeloid leukemia K-562 (ATCC: CRL 3344, Middlesex, UK), and the human acute lymphocytic leukemia REH (ATCC: CRL 8286, Middlesex, UK) were provided by UIM en Genética Humana del Hospital de Pediatría de CMN S XXI, IMSS. Cell cultures were tested for mycoplasma contamination using MycoAlert mycoplasma detection kit (Lonza Walkersville, Inc., Walkersville, MD, USA). The cell lines were cultured in RPMI 1640 medium containing 10% (*v*/*v*) fetal bovine serum and maintained at 0.5−1.0 × 10^6^ cell mL^−1^ in a humidified atmosphere containing 5% CO_2_ at 37 °C. Cell viability was determined by the trypan blue exclusion test. Cells were resuspended in fresh medium 24 h before treatments to ensure the exponential growth. 

#### 4.2.2. Cytotoxic Activity

The cytotoxic activity of incomptines A (**IA**) and B (**IB**) against leukemia human cell lines was assessed using the colorimetric 3-(4,5-dimethyl-2-thiazolyl)-2,5-diphenyl-tetrazolium bromide (MTT) test. Exponentially growing cells of U-937, HL-60, K-562, and REH cell lines were seeds in 96-well plates at a density of 5.0 x 10^3^ cells per well in 100 µL and were treated with five serial concentrations between 0.1 µM and 5.0 µM of HTSLs or etoposide or methotrexate for 24 h under 5% CO_2_ and 95% O_2_ at 37 °C. The compounds were dissolved in DMSO; the final concentration of DMSO used was 0.1% (*v*/*v*) for each sample. Cells (U-937 or HL-60 or K-562 or REH) treated with 0.1% DMSO served as the control group. After incubation for specified times, MTT reagent (10 µL, 5 mg dissolved in 1 mL of PBS) was added to each well and incubated for 4 h. The plates were centrifuged (10 min at 350 × g) and the purple formazan crystals of metabolized yellow tetrazolium salt by viable cells were dissolved in 150 µL of DMSO. Absorbance was quantified at 570 nm using the ELISA plate reader. Results were expressed as a percentage of viability, with 100% representing control cells treated with 0.1% DMSO alone. Then, the CC_50_ was determined. This was defined as the treatment concentration at which a 50% reduction in cellular proliferation was observed. This was calculated graphically using the curve-fitting algorithm of the computer software Prism 5.03 (GraphPad, La Jolla, CA, USA). Values were calculated as means ± S.E.M from three independent experiments, each performed in triplicate. 

### 4.3. Molecular Docking of Incomptine A (**IA**), Incomptine B (**IB**), Etoposide (**ET**), and Methotrexate (**MTX**) 

Incomptine A (**IA**), incomptine B (**IB**), etoposide (**ET**), and methotrexate (**MTX**) structure was created and prepared using MOE software [28]. Several three-dimensional structures involved in treating cancers were retrieved from the Protein Data Bank (https://www.rcsb.org, accessed on 16 February 2022) with the following access codes: human topoisomerase IIα (PDB id: 5GWK, Resolution: 3.15 Å), human topoisomerase IIβ (PDB id: 3QX3, Resolution: 2.16 Å), human dihydrofolate reductase (PDB id: 3EIG, Resolution: 1.70 Å), human methylenetetrahydrofolate dehydrogenase (PDB id: 6ECQ, Resolution: 2.70 Å), and human B-cell lymphoma 2 protein (PDB id: 4LVT, Resolution: 2.05 Å). Molecular targets and ligands (incomptine A, incomptine B, etoposide, and methotrexate) were submitted to MOE software. All water molecules and cocrystal ligands were removed from the crystallographic structures. Then, each one of the hydrogen atoms was added, non-polar hydrogen atoms were merged, and Gasteiger charges were assigned to all molecules (ligands and proteins). Next, the torsions from compounds were allowed to rotate during the docking study. The molecular docking experiments were carried out using AutoDock Vina with 20 modes and an exhaustiveness value of 16 [29,30]. The active site of each target was covered with the proper size of the grid. The grid was centered at the following coordinates: (center_x = 23.879; center_y = −38.695 and center_z = 39.672; for 5GWK), (center_x = 27.837; center_y = 102.863 and center_z = 36.588; for 3QX3), (center_x = 10.545; center_y = −6.569 and center_z = −18.125; for 3EIG), (center_x = 1.655; center_y = 57.192 and center_z = 19.537; for 6ECQ) and (center_x = 7.154; center_y = −3.142 and center_z = −7.712; for 4LVT). Size: (20 × 20 × 20 points, for 5GWK), (30 × 30 × 30 points, for 3QX3), (20 × 20 × 20 points, for 3EIG), (20 × 26 × 20 points, for 6ECQ), and (20 × 28 × 18 points, for 4LVT) with a default spacing.

#### Molecular Docking Validation

The molecular docking protocol was validated through a re-docking of co-crystal ligands into the binding site of both pharmacological targets. The conditions to reproduce the binding mode of co-crystallographic ligands were established after re-docking; we found that the root-mean-square deviations (RMSDs) between the co-crystal ligands and the re-docked structures were less than 2.0 Å, for all targets. These conditions allowed us to obtain good predictions in the compounds of interest.

### 4.4. Statistical Analysis

Inhibition percentage was plotted against concentration; the best straight line was determined by regression analysis, and the CC_50_ was calculated. All data were expressed as mean ± standard deviation of nine measurements. Statistical analysis of data was performed using one-way ANOVA. A maximum probability value of *p* < 0.05 was considered statistically significant and pairwise differences were analyzed by Bonferroni post hoc test. Analyses were performed using GraphPad Prism Version 5.03 (GraphPad Software Inc., La Jolla, CA, USA).

## 5. Conclusions

The heliangolide-type sesquiterpene lactones, incomptines A (**IA**) and B (**IB**), isolated from the dichloromethane extract of aerial parts of *Decachaeta incompta*, showed significant cytotoxic activity against four leukemia human cell lines: U-937, HL-60, K-562, and REH cells. In addition, molecular docking studies suggest that cytotoxic effects may be explained by the affinity of these compounds for the five proteins used as targets and associated with different cancer processes including topoisomerase IIα (TIIα), topoisomerase IIβ (TIIβ) dihydrofolate reductase (DHFR), tetrahydrofolate synthase (MTHFD), and Bcl-2-related protein A1 (BCL-2). Although more research must be carried out to discover how both HTSLs cause cell death, these compounds have pharmacological potential as anticancer agents, specifically against leukemia. Both HTSLs, **IA** and **IB**, should be considered for additional preclinical and in vivo studies. 

## Figures and Tables

**Figure 1 molecules-27-01687-f001:**
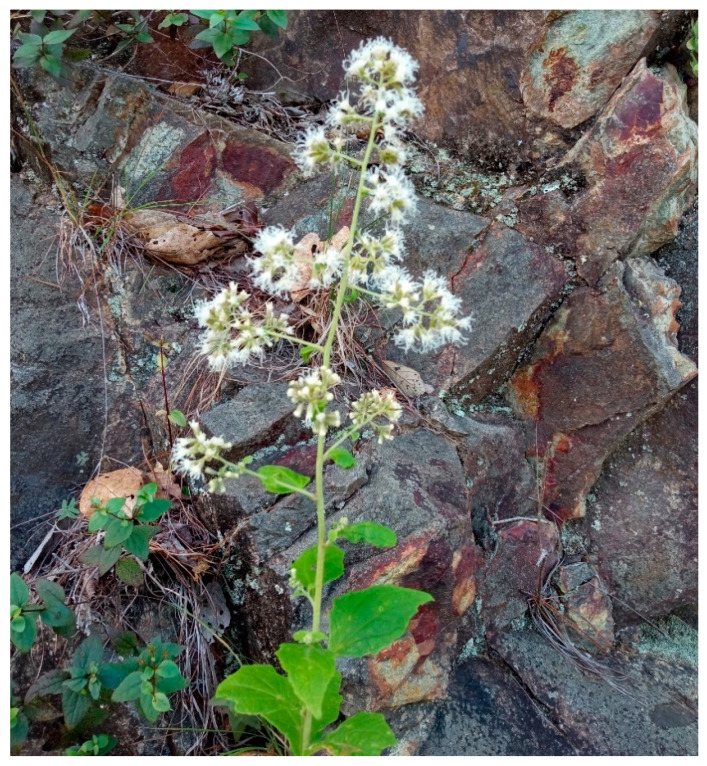
*Decachaeta incompta* (DC) R. M. King and H. Robinson.

**Figure 2 molecules-27-01687-f002:**
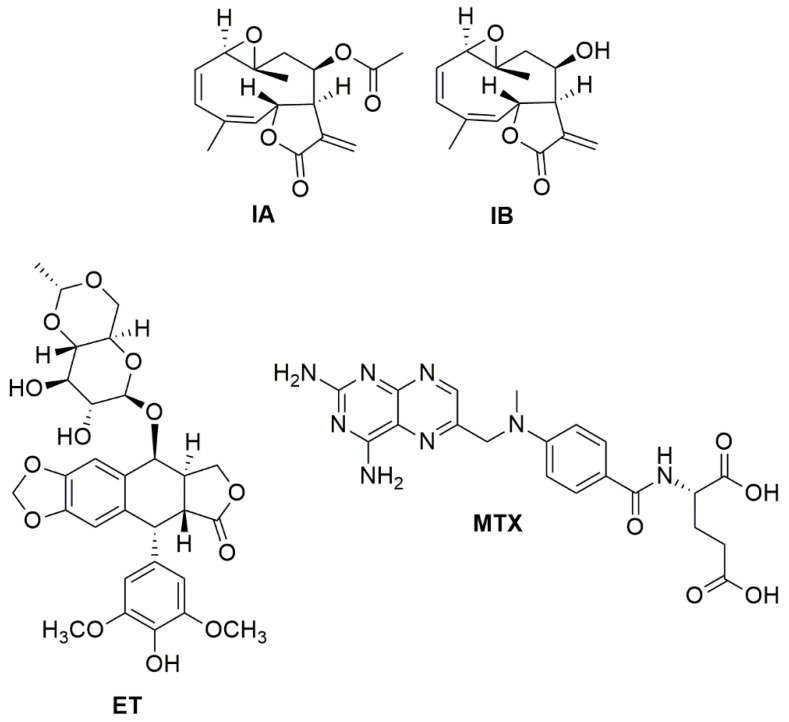
Structures of incomptine A (**IA**), incomptine B (**IB**), methotrexate (**MTX**), and etoposide (**ET**).

**Figure 3 molecules-27-01687-f003:**
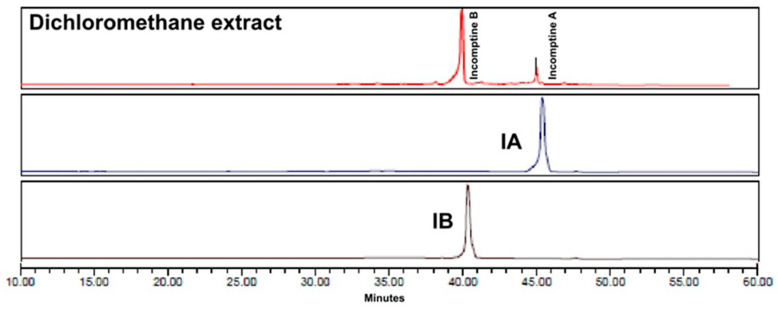
High-performance liquid chromatography with diode-array detection (HPLC-DAD) analysis at 240 nm of incomptine A (gray), incomptine B (brown), and dicholorometane extract from *Decachaeta incompta* (red).

**Figure 4 molecules-27-01687-f004:**
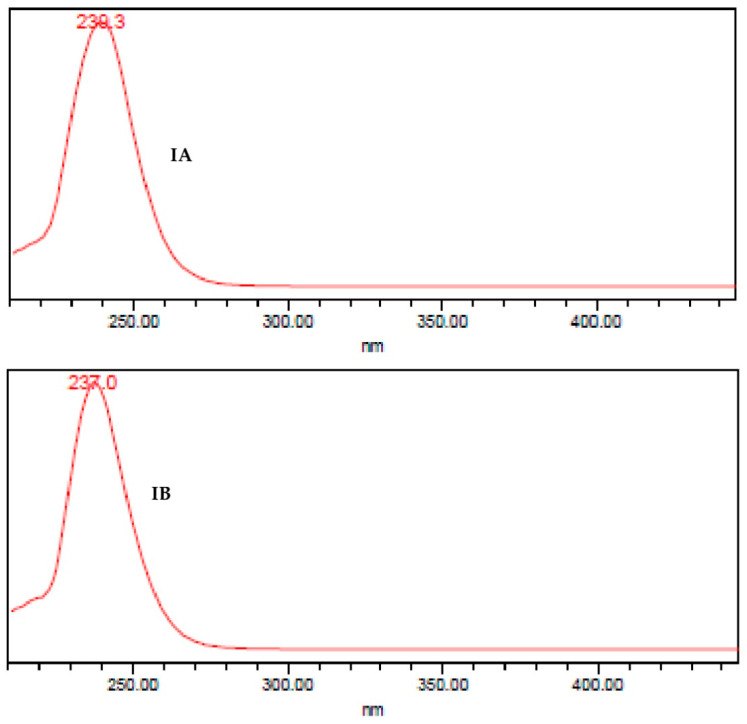
UV spectrum obtained of HPLC-DAD analysis of incomptine A (**IA**) and incomptine B (**IB**).

**Figure 5 molecules-27-01687-f005:**
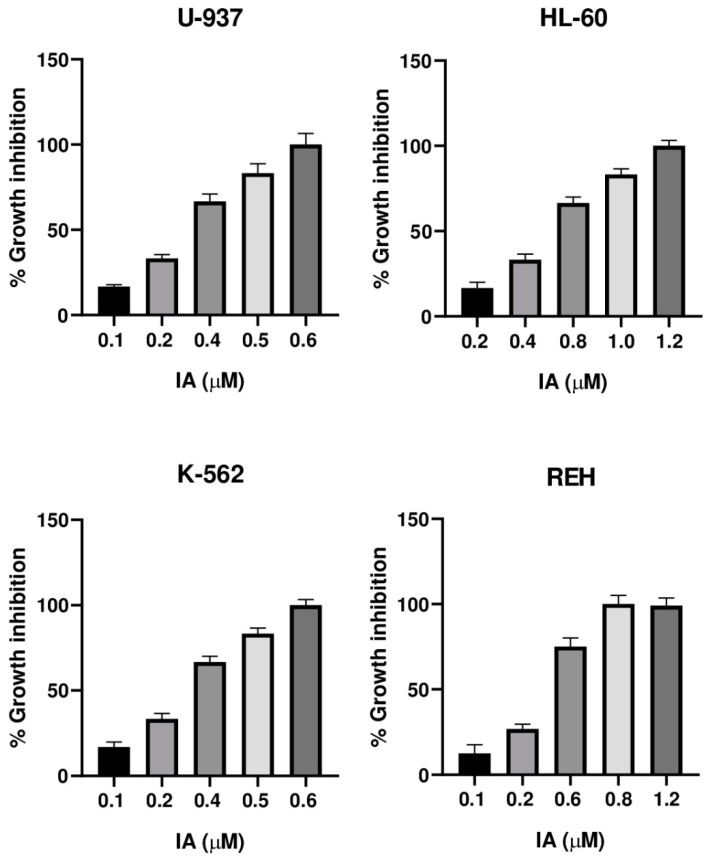
Cytotoxicity of incomptine A (**IA**) against U-937, HL-60, K-562, and REH cell lines.

**Figure 6 molecules-27-01687-f006:**
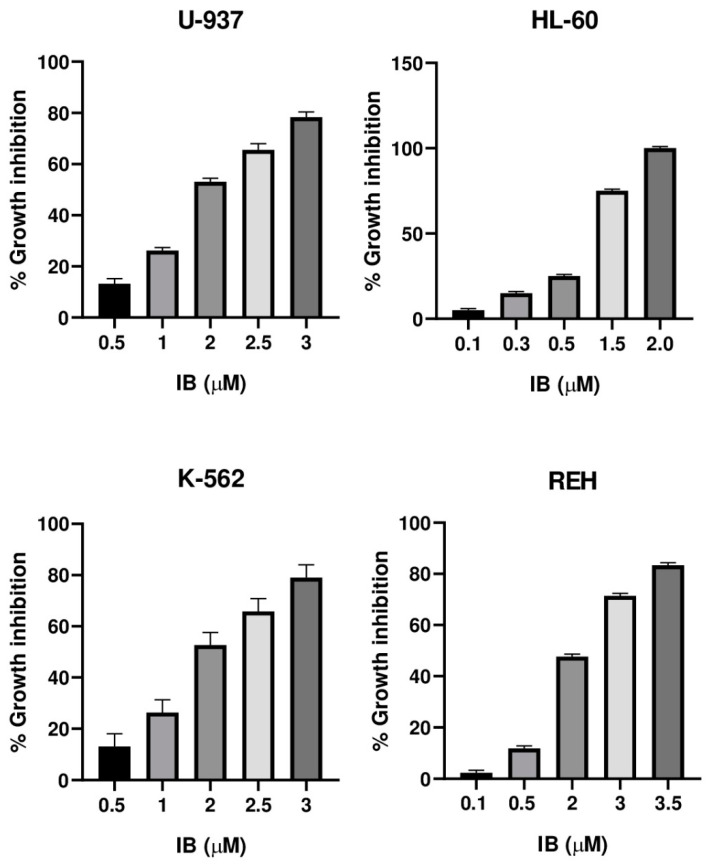
Cytotoxicity of incomptine B (**IB**) against U-937, HL-60, K-562, and REH cell lines.

**Figure 7 molecules-27-01687-f007:**
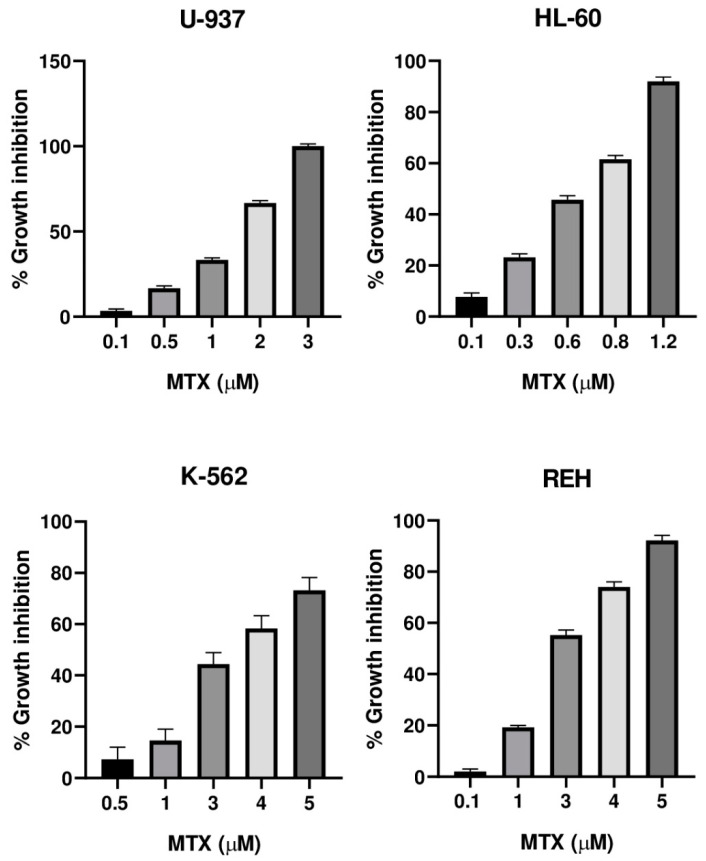
Cytotoxicity of methotrexate (**MTX**) against U-937, HL-60, K-562, and REH cell lines.

**Figure 8 molecules-27-01687-f008:**
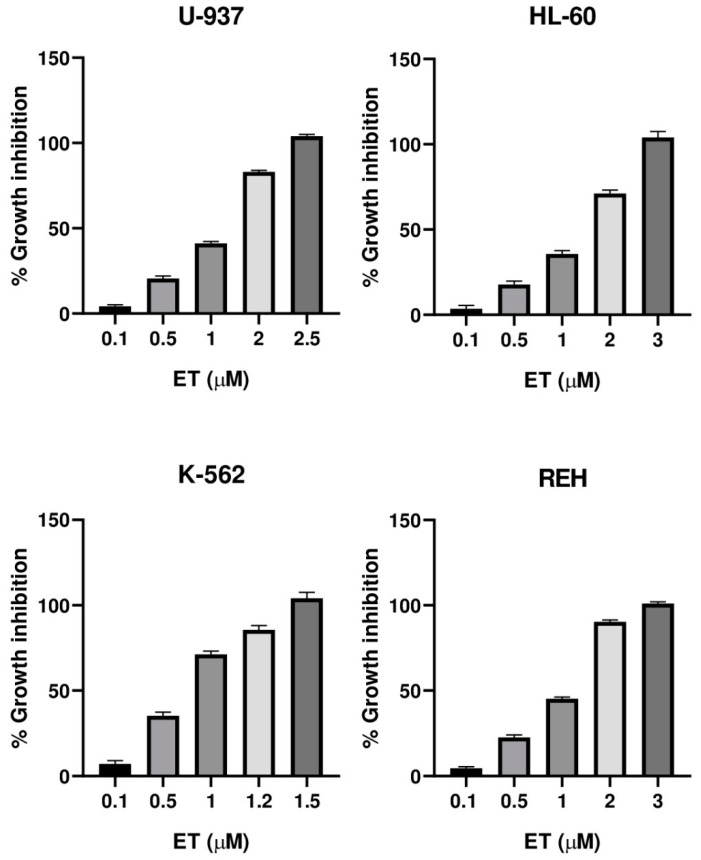
Cytotoxicity of etoposide (**ET**) against U-937, HL-60, K-562, and REH cell lines.

**Figure 9 molecules-27-01687-f009:**
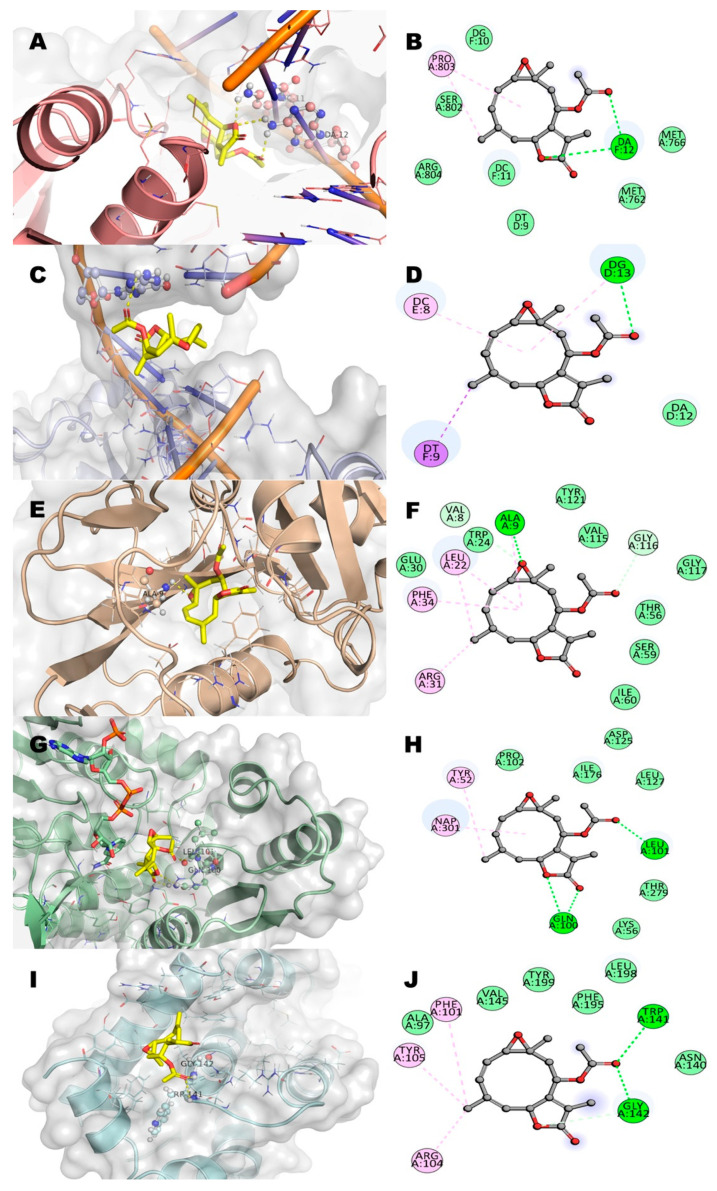
2D and 3D interaction of incomptine A (**IA**, yellow color) with residues in the binding site of several targets in cancer. (**A**,**B**) Human topoisomerase IIα (TIIα); (**C**,**D**) human topoisomerase IIβ (TIIβ); (**E**,**F**) human dihydrofolate reductase (DHFR); (**G**,**H**) human methylenetetrahydrofolate dehydrogenase (MTHFD); and (**I**,**J**) human B-cell lymphoma 2 protein (BCL2).

**Figure 10 molecules-27-01687-f010:**
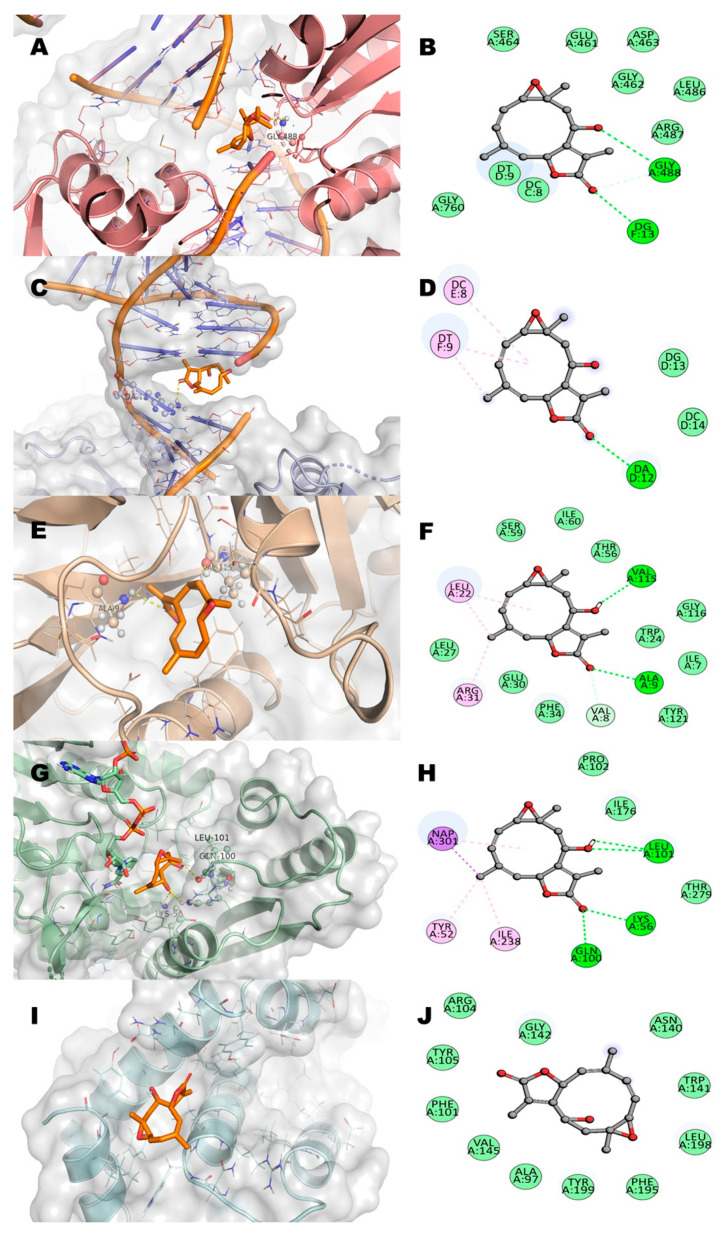
2D and 3D interaction of incomptine B (**IB**, orange color) with residues in the binding site of several targets in cancer. (**A**,**B**) Human topoisomerase IIα (TIIα); (**C**,**D**) human topoisomerase IIβ (TIIβ); (**E**,**F**) human dihydrofolate reductase (DHFR); (**G**,**H**) human methylenetetrahydrofolate dehydrogenase (MTHFD); and (**I**,**J**) human B-cell lymphoma 2 protein (BCL2).

**Figure 11 molecules-27-01687-f011:**
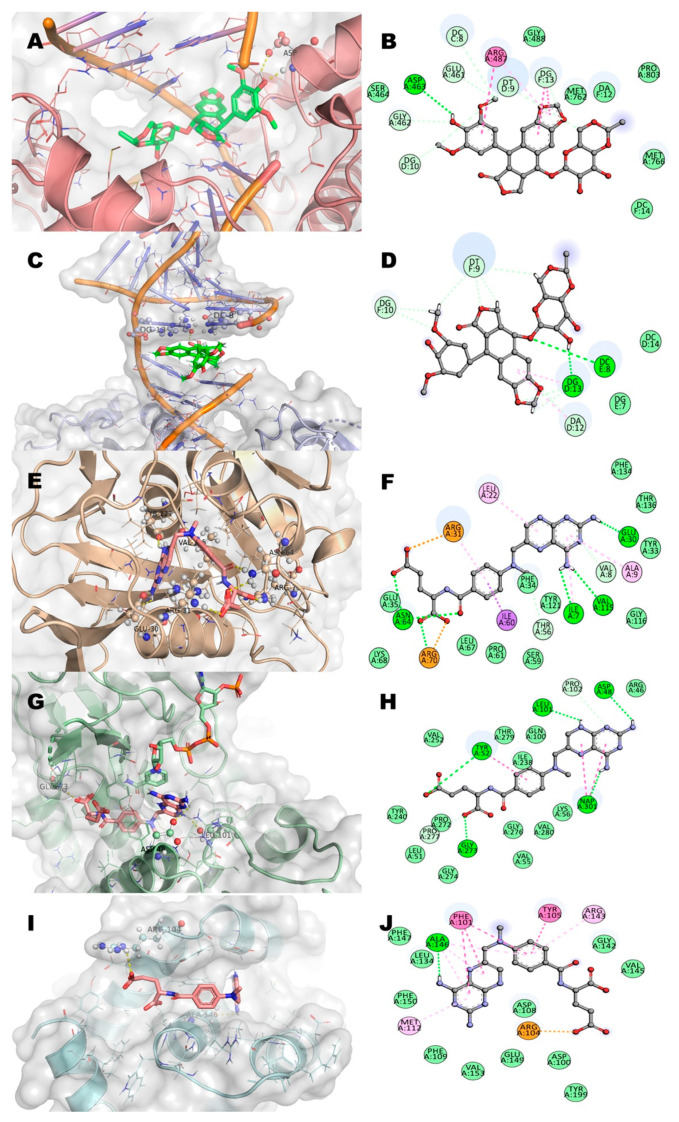
2D and 3D interaction of etoposide (**ET**, green color) and methotrexate (**MTX**, pink color) with residues in the binding site of several targets in cancer. (**A**,**B**) Human topoisomerase IIα (TIIα); (**C**,**D**) human topoisomerase IIβ (TIIβ); (**E**,**F**) human dihydrofolate reductase (DHFR); (**G**,**H**) human methylenetetrahydrofolate dehydrogenase (MTHFD); and (**I**,**J**) human B-cell lymphoma 2 protein (BCL2).

**Table 1 molecules-27-01687-t001:** Cytotoxic activities of incomptines A (**IA**) and B (**IB**) isolated from dichloromethane extract of the aerial parts of *Decachaeta incompta*.

Sample	Leukemia and Lymphoma Cell Lines (CC_50_ µM) ^a^			
	U-937	HL-60	K-562	REH
Incomptine A (**IA**)	0.3 ± 0.02	0.6 ± 0.02	0.3 ± 0.01	0.4 ± 0.02
Incomptine B (**IB**)	1.9 ± 0.2	1.0 ± 0.1	1.9 ± 0.1	2.1 ± 0.03
Methotrexate (**MTX**)	1.5 ± 0.02	0.65 ± 0.01	3.4 ± 0.2	2.7 ± 0.02
Etoposide (**ET**)	1.2 ± 0.01	1.4 ± 0.03	0.7 ± 0.02	1.1 ± 0.01

^a^ U-937 (histiocytic lymphoma), HL-60 (acute promyelocytic leukemia), K-562 (chronic myeloid leukemia), and REH (acute lymphocytic leukemia); CC_50_ was defined as the treatment concentration at which 50% reduction in cellular proliferation was observed. Data were analyzed using Graph Pad Prism, (*n* = 3), *p* < 0.05. This was calculated graphically using the curve-fitting algorithm of the computer software Prism 5.03 (GraphPad, La Jolla, CA, USA). Values were calculated as means ± S.E.M from three independent experiments, each performed in triplicate.

**Table 2 molecules-27-01687-t002:** ∆G (kcal/mol) and receptor–ligand interactions from molecular docking.

Compound	Incomptine A (IA)	Incomptine B (IB)	Methotrexate (MTX)	Etoposide (ET)
∆G	−7.1	−7.4	-	−12.3
TIIα HBR	DA12	Gly488, DG13	-	Asp463
NPI	Glu461, Ser464, Arg487	Glu461, Ser464, Arg487	-	Glu461, Gly462, Arg487
∆G	−6.5	−5.7	-	−9.2
TIIβ HBR	DG13	DA12	-	DC8, DG13
NPI	DC8, DT9, DA12	DC8, DT9, DG13	-	DA12, DT9, DG10
∆G	−7.8	−8.1	−9.6	-
DHFR HBR	Ala9	Ala9, Val115	Ile7, Glu30, Arg31, Asn64	-
NPI	Leu22, Trp24, Arg31, Phe34, Thr56, Ser59, Ile60, Val115, Gly116, Tyr121	Leu22, Trp24 Leu27, Glu30, Arg31, Phe34, Thr56, Ser59, Ile60, Gly116, Tyr121	Val8, Ala9, Leu22, Phe34, Glu35, Thr56, Ser59, Ile60, Pro61, Leu67, Phe134	-
∆G	−7.6	−7.9	−9.1	-
THFS HBR	Gln100, Leu101	Gln100, Leu101, Lys56	Asp48, Tyr53, Leu101, Gly273	-
NPI	NADP, Tyr52, Leu56, Pro102, Ile176, Thr279	NADP, Tyr52, Ile176, Ile238, Thr279	NADP, Lys56, Gln100, Pro102, Ile138, Tyr240, Pro272, Gly276	-
∆G	−7.3	−7.3	−8.1	-
BCL2 HBR	Trp141, Gly142	-	Arg104, Ala146	-
NPI	Ala97, Phe101, Arg104, Tyr105, Asn140, Val145, Phe195, Tyr199	Ala97, Gln100, Phe101, Arg104, Tyr105, Asn140, Trp141, Gly142, Val145, Phe195, Leu198, Tyr199	Phe101, Tyr105, Asp108, Met112, Leu134, Gly142, Arg143, Val145, Phe150	-

Asp: aspartate; Asn: asparagine; Arg: arginine; Gln: glutamine; Lys: lysine; Thr: threonine; Ser: serine; Trp: tryptophan; Leu: leucine; Gly: glycine; Glu: glutamic acid; Ile: isoleucine; Phe: phenylalanine; Pro: proline; Val: valine; DA: deoxyadenosine; DG: deoxyguanosine; DT: deoxythymidine; DC: deoxycytidine; NADP: nicotinamide adenine dinucleotide phosphate; ΔG: binding energy; HBR: H-bonding residues; NPI: nonpolar interactions; TIIα: topoisomerase IIα; TIIβ: topoisomerase IIβ; DHFR: dihydrofolate reductase; BCL2: B-cell lymphoma 2 protein.

## Data Availability

The data presented or additional data on this study are available on request from corresponding author.

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
