# Peer review of "Expanding the Study of the Cytotoxicity of Incomptines A and B against Leukemia Cells"

_molecules, 2022, doi:10.3390/molecules27051687_

Round 1

Reviewer 1 Report

The manuscript entitled "Additional Cytotoxic Activity of Incomptines A and B Against Human Cancer Cell Lines" reported the cytotoxic activity of two heliangolide-type sesquiterpene lactones on four human cancer cell lines.  The paper is well structure and written (minor spell check required). However, from my point of view, the manuscript has several major flaws:

  • Originality is very low.
    • The authors evaluated the cytotoxic activity of only two sesquiterpenes on four leukemia cell lines. The authors have previously shown the cytotoxic activity of IA on lymphoma cells. Although leukemias and lymphomas are different types of cancers, since they showed cytotoxicity in lymphoma cells, it could be expected that they are cytotoxic in leukemia cells. In fact, they tested on U-937 cells before and considered this cell line as lymphoma cells, but they considered these cells as leukemia cells in this manuscript. In my opinion, it would have been more interesting to evaluate the selective anticancer activity of these SLs, treating cancer cells and nonmalignant cells. If they have potential anticancer activity, they should kill cancer cells at concentrations that do not significantly affect nonmalignant cells.
    • The interaction of incomptine A with the Human B-cell lymphoma 2 protein (BCL2) has previously been reported (Calzada et al. 2021).

Calzada F, Bautista E, Hidalgo-Figueroa S, García-Hernández N, Barbosa E, Velázquez C, Ordoñez-Razo RM, Arietta-García AG. Antilymphoma Effect of Incomptine A: In Vivo, In Silico, and Toxicological Studies. Molecules. 2021; 26(21):6646. https://doi.org/10.3390/molecules26216646

Pina-Jiménez E, Calzada F, Bautista E, Ordoñez-Razo RM, Velázquez C, Barbosa E, García-Hernández N. Incomptine A Induces Apoptosis, ROS Production and a Differential Protein Expression on Non-Hodgkin’s Lymphoma Cells. International Journal of Molecular Sciences. 2021; 22(19):10516. https://doi.org/10.3390/ijms221910516

  • The authors should show the cell viability curves (dose-response curves) from the MTT assay for a better understanding of the results.

Minor revision:

  • From my point of view, the title is not appropriate. Authors should specify type of cancer cells used in the title and abstract.
  • It is necessary to check the use of abbreviations. For example, SL is not defined until line 230.
  • The authors should better define the statistical analysis in Table 1.
  • I think they should change the CC50 definition. The authors defined it as 'Cytotoxic concentration required to kill 50% of cells'. The MTT assay is a detection of cell proliferation, it does not distinguish whether loss of proliferation is due to cell death or growth arrest.
  • I recommend using the same units for concentrations of compounds. The authors used microM in the results and discussion, but microg/mL in Methods (line 319).
  • Minor spell check required. For example:
    • “IA and IB has cytotoxic activity…” line 33
    • “… and tretaed with five serial concentrations…” line 319

However, the molecular docking analysis is very interesting. Because these SLs showed an interaction similar to that of etoposide and methotrexate, these lactones could induce DNA damage. Authors should explore this possible mechanism of action in the future.

Author Response

Reviewer1

Comments and Suggestions for Authors

The manuscript entitled "Additional Cytotoxic Activity of Incomptines A and B Against Human Cancer Cell Lines" reported the cytotoxic activity of two heliangolide-type sesquiterpene lactones on four human cancer cell lines.  The paper is well structure and written (minor spell check required). However, from my point of view, the manuscript has several major flaws:

  • Originality is very low.
    • The authors evaluated the cytotoxic activity of only two sesquiterpenes on four leukemia cell lines. The authors have previously shown the cytotoxic activity of IA on lymphoma cells. Although leukemias and lymphomas are different types of cancers, since they showed cytotoxicity in lymphoma cells, it could be expected that they are cytotoxic in leukemia cells. In fact, they tested on U-937 cells before and considered this cell line as lymphoma cells, but they considered these cells as leukemia cells in this manuscript. In my opinion, it would have been more interesting to evaluate the selective anticancer activity of these SLs, treating cancer cells and nonmalignant cells. If they have potential anticancer activity, they should kill cancer cells at concentrations that do not significantly affect nonmalignant cells.
    • The interaction of incomptine A with the Human B-cell lymphoma 2 protein (BCL2) has previously been reported (Calzada et al. 2021).

Calzada F, Bautista E, Hidalgo-Figueroa S, García-Hernández N, Barbosa E, Velázquez C, Ordoñez-Razo RM, Arietta-García AG. Antilymphoma Effect of Incomptine A: In Vivo, In Silico, and Toxicological Studies. Molecules. 2021; 26(21):6646. https://doi.org/10.3390/molecules26216646

Pina-Jiménez E, Calzada F, Bautista E, Ordoñez-Razo RM, Velázquez C, Barbosa E, García-Hernández N. Incomptine A Induces Apoptosis, ROS Production and a Differential Protein Expression on Non-Hodgkin’s Lymphoma Cells. International Journal of Molecular Sciences. 2021; 22(19):10516. https://doi.org/10.3390/ijms221910516

  • The authors should show the cell viability curves (dose-response curves) from the MTT assay for a better understanding of the results.

Answer to Comments:

Dear reviewer thanks a lot for your comments and suggestion. In this sense the current manuscript is a part on anticancer potencial of incomptines. Is true that antilymphoma properties of incomptine A has been reported using in vivo, in vitro and in silico approachs. In this manuscript really Incomptine A is used as comparative natural product to know the antitumor potential of incomptine B. Therefore, was used U-937 cells and as target BCL2 in the docking method. Our results suggest. 1.- antitumor potential of incomptine B. 2.- Expanded the antitumor properties of incomptine A on other cancer cells. 3.- Give additional support to antitumor potential of HTSLs. 

Also, dose-response curves of IA against all cell lines used were included in Results as Figure 5.

Minor revision:

Query 1

  • From my point of view, the title is not appropriate. Authors should specify type of cancer cells used in the title and abstract.

Answer:

The title was changed as:

Expanding the Study of the Cytotoxicity of Incomptines A and B Against Leukemia Cells.

Also, the abstract was modificated.

Query 2:

  • It is necessary to check the use of abbreviations. For example, SL is not defined until line 230.

Answer.

The text was checked and abbreviation SLs was defined in the introduction, line 55.

Query 3

  • The authors should better define the statistical analysis in Table 1.

Answer:

Additional information was included in Table 1

Query4

  • I think they should change the CC50 definition. The authors defined it as 'Cytotoxic concentration required to kill 50% of cells'. The MTT assay is a detection of cell proliferation, it does not distinguish whether loss of proliferation is due to cell death or growth arrest.

Answer:

CC50 was defined as: the treatment concentration at which 50% reduction in cellular proliferation was observed.

Query5

  • I recommend using the same units for concentrations of compounds. The authors used microM in the results and discussion, but microg/mL in Methods (line 319).

Answer:

Units were corrected in Methods as “serial concentrations between 0.1 μM and 4.0 μM”

Query6

  • Minor spell check required. For example:
    • “IA and IB has cytotoxic activity…” line 33
    • “… and tretaed with five serial concentrations…” line 319

However, the molecular docking analysis is very interesting. Because these SLs showed an interaction similar to that of etoposide and methotrexate, these lactones could induce DNA damage. Authors should explore this possible mechanism of action in the future.

Answer:

-Although three reviewers’ comment that “Minor spell check required” the manuscript was checked by all authors and a
native English-speaking colleague.

-Line 33 was checked “have”

-Line 319 was checked “treated”

-In agreement with reviewer in future works on DNA damage will be explored

Reviewer 2 Report

Sesquiterpene lactones are secondary metabolites that have several biological properties such  as cytotoxic, antitumor, antimicrobial and antiinflammatory. Herein, the biological properties of SLs have been associated with the presence of an a-methylene-γ−lactone in the structure of these class of metabolites, this moiety has capacity to act as a Michel acceptor and react with sulfhydryl residues of proteins. In this study, the Authors investigated the cytotoxic activity (U-937, HL-60, K-562, and REH) and docking analysis on five molecular targets with relevance in cancer treatment of the two major heliangolide-type sesquiterpene lactones from Decachaeta incompta. Results showed that incomptine A exhibited the best cytotoxic activity, its effects were best than methotrexate and etoposide, two antitumor agents used currently for the treatment of cancer and used as positive  controls. Moreover, it is important to note that although incomptine B was less potent than A, its cytotoxic activity was close than etoposide and methotrexate, suggesting that also it is good candidate for the development on new anticancer drugs. Therefore I appreciate work done but manuscript’s Authors should make the some additions and corrections.

Section “Results”:

  • Poor quality of Figure 2. Please prepare the chemical structures in the appropriate graphics program.
  • Figure 3 and Figure 3, here we have unreadable chromatograms. Please correct their resolution.

Section “Materials and methods”

  • Line 274 and 275: Please replace the MeOH abbreviation with its full name i.e. methanol (MeOH).
  • If we do not use chromatographic standards, it is worth using the LC-MS or NMR technique to be sure about the identification of the tested compounds.

Best regards

Author Response

Reviewer 2

Comments and Suggestions for Authors

Section “Results”:

Query 1:

  • Poor quality of Figure 2. Please prepare the chemical structures in the appropriate graphics program.

Answer:

The Figure 3 was working to obtain the best quality

Query2

  • Figure 3 and Figure 3, here we have unreadable chromatograms. Please correct their resolution.

Answer:

Structures were drawn using the ChemDraw 12.0 software and applying the template ACS 1996. For inclusion of the structures in the text, these were directly embedded.

Section “Materials and methods”

In addition, figures 5-7 were working to obtain best quality 

Query3

  • Line 274 and 275: Please replace the MeOH abbreviation with its full name i.e. methanol (MeOH).

Answer:

All manuscript was checked and MeOH abbreviation was replaced by methanol

Query4

  • If we do not use chromatographic standards, it is worth using the LC-MS or NMR technique to be sure about the identification of the tested compounds.

Answer:

Dear reviewer NMR, was used to identification of both incomptines. NMR-H1 and C13 were recorded to both incomptines. Additional text line 365 was included (and NMR-H1 and C13). The characterization and full assignation were reported previously therefore is unnecessary to be included in this paper.

Reviewer 3 Report

The research is relevant, since the sesquiterpene lactones of the heliangolide type, incomptins A (IA) and B (IB) isolated
of the dichloromethane extract of aerial parts of Decachaeta incompta showed
pharmacological potential as anticancer agents specifically against leukemia.

Recommendations

Standardize font color on lines 67-72.

Author Response

Reviewer 3

Comments and Suggestions for Authors

Query:

Recommendations

Standardize font color on lines 67-72.

Answer:

Font color on lines 67-72 was standardized

Reviewer 4 Report

The manuscript entitled “Additional cytotoxic activity of incomptines A and B against human cancer cell lines” is so well structured and written, the conclusions are supported by the data presented in the section of Results therefore your manuscript can be accepted for publication in this Journal, Molecules. However, it is still having some comments for revision below

  1. To be sure the cytotoxic activity of the incomptines A and B isolated of dichloromethane extract of the aerial parts from Decachaeta incompta in leukemia cells showed in Table 1, it is required to provide the results of Annexin V/PI and western blot of cleaved PARP and cleaved Caspase-3 in leukemia cells treated of incomptines A and B.
  2. To confirm the cytotoxic activity of the incomptines A and B in leukemia cells. It is required to explain the molecular mechanism of cytotoxic activity by the incomptines A and B in these cells.
  3. The number of keywords is not enough and significant therefore it is required to add more significant keywords.
  4. The number of cited reference is small therefore please pay an attention to add/provide more the cited references. Furthermore, some sections of your manuscript as Introduction, Discussion and so on should be rewritten after adding more the cite of references.
  5. Some sections of your manuscript as Introduction and Discussion are poor in English and preparation.

Author Response

Reviewer 4

Comments and Suggestions for Authors

The manuscript entitled “Additional cytotoxic activity of incomptines A and B against human cancer cell lines” is so well structured and written, the conclusions are supported by the data presented in the section of Results therefore your manuscript can be accepted for publication in this Journal, Molecules. However, it is still having some comments for revision below

Query1

  1. To be sure the cytotoxic activity of the incomptines A and B isolated of dichloromethane extract of the aerial parts from Decachaeta incompta in leukemia cells showed in Table 1, it is required to provide the results of Annexin V/PI and western blot of cleaved PARP and cleaved Caspase-3 in leukemia cells treated of incomptines A and B.

Answer to comments:

In future Works on antitumor properties of incomptines A and B their effects will be explored including your suggestion as Annexin V/PI and western blot of cleaved PARP and cleaved Caspase-3. Also, proteomic studies are in process.  However, the pandemic situation of Covid19 the cytometer of flow and other equipment and reactive have a problem.  We believe that results could be in 6 months or one year. 

Query2

  1. To confirm the cytotoxic activity of the incomptines A and B in leukemia cells. It is required to explain the molecular mechanism of cytotoxic activity by the incomptines A and B in these cells.

Answer:

In silico results suggest potential effects on specific targets in cancer cells used.

Additional text was included:

Also, the molecular docking analysis suggest that cytotoxic activity of IA and IB against all leukemia cell lines used in this study may be associated with the effects on the five targets used: human topoisomerase IIa, human topoisomerase IIβ, human dihydrofolate reductase, human methylenetetrahydrofolate dehydrogenase and human B-cell lymphoma 2 protein.

Also, additional studies are in process to confirm of mechanism of cytotoxic activity. We believe to obtain results in 6 months or one year.

Query3

  1. The number of keywords is not enough and significant therefore it is required to add more significant keywords.

Answer:

Keywords was modificated as:

incomptine A; incomptine B, sesquiterpene lactones; cytotoxic activity; human cancer cell lines, leukemia; docking

Query 4 and 5

  1. The number of cited reference is small therefore please pay an attention to add/provide more the cited references. Furthermore, some sections of your manuscript as Introduction, Discussion and so on should be rewritten after adding more the cite of references.
  2. Some sections of your manuscript as Introduction and Discussion are poor in English and preparation.

Answer 4 and 5

- Additional references were included and introduction and discussion were rewritten.

-Although 3 reviewer report “fine/minor spell check is required” the manuscript was checked by all authors and by a
native English-speaking colleague.

Round 2

Reviewer 1 Report

The authors replied to almost all my queries.

However, the authors should show dose-response curves for all compounds (Incomptine A, Incomptine B, Methotrexate and Etoposide), at least as a supplementary figure if they do not want to show them as a main figure. The graphs should show mean and SEM bars.

The authors should consider evaluating the selective anticancer activity of these SLs, treating cancer cells and non-malignant cells, in the future.

English language and style are fine. 

Author Response

Reviewer 1

Comments and Suggestions for Authors

The authors replied to almost all my queries.

Query1.

However, the authors should show dose-response curves for all compounds (Incomptine A, Incomptine B, Methotrexate and Etoposide), at least as a supplementary figure if they do not want to show them as a main figure. The graphs should show mean and SEM bars.

Answer:

Dose-response curves for all compounds (Incomptine A, Incomptine B, Methotrexate and Etoposide) were included in the manuscript including mean and SEM bars

Query2

The authors should consider evaluating the selective anticancer activity of these SLs, treating cancer cells and non-malignant cells, in the future.

 Answer:

In the future we consider evaluating the selective anticancer activity of these SLs, treating cancer cells.

Query 3

English language and style are fine. 

Answer:

-Although three reviewers’ comment that “English language and style are fine” the manuscript was checked by all authors and a native English-speaking colleague.

Reviewer 4 Report

The revised version of your manuscript entitled “Expanding the study of the cytotoxicity of incomptines A and B against leukemia cells” can be accepted for publication in this Journal, Molecules.

Author Response

Reviewer 4

Comments and Suggestions for Authors

The revised version of your manuscript entitled “Expanding the study of the cytotoxicity of incomptines A and B against leukemia cells” can be accepted for publication in this Journal, Molecules.

Answer:

Don’t query was suggest

Thanks a lot for your positive answer
